# A qualitative exploration of cardiovascular disease patients' views and experiences with an eHealth cardiac rehabilitation intervention: The PATHway Project

Orlagh O'Shea[1]*, Catherine Woods[2], Lauri McDermott[3], Roselien Buys[4], Nils Cornelis[5], Jomme Claes[4], Véronique Cornelissen[5], Anne Gallagher[6], Helen Newton[7], Niall Moyna[8], Noel McCaffrey[3], Davide Susta[8], Clare McDermott[8], Ciara McCormack[8], Werner Budts[4], Kieran Moran[8,9]

1 School of Physiotherapy, Royal College of Surgeons of Ireland, Dublin, Ireland, 2 Department of Physical Education and Sport Sciences, Physical Activity for Health, Health Research Institute, Faculty of Education and Health Sciences, University of Limerick, Limerick, Ireland, 3 ExWell Medical, Dublin, Ireland, 4 Department of Cardiovascular Sciences, KU Leuven, Leuven, Belgium, 5 Department of Rehabilitation Sciences, KU Leuven, Leuven, Belgium, 6 Mater Misericordiae University Hospital, Dublin, Ireland, 7 Beaumont University Hospital, Dublin, Ireland, 8 Department of Health & Human Performance, Dublin City University, Dublin, Ireland, 9 Insight Centre for Data Analytics, Dublin City University, Dublin, Ireland

* orlaghoshea@rcsi.com

## Abstract

The aim of this study is to explore participants' views and experiences of an eHealth phase 3 cardiac rehabilitation (CR) intervention: Physical Activity Towards Health (PATHway). Sixty participants took part in the PATHway intervention. Debriefs were conducted after the six-month intervention. All interviews were audio recorded and transcribed verbatim. Transcripts were analysed with Braun and Clarke's thematic analysis. Forty-four (71%) debriefs were conducted (n = 34 male, mean (SD) age 61 (10) years). Five key themes were identified: (1) Feedback on the components of the PATHway system, (2) Motivation, (3) Barriers to using PATHway, (4) Enablers to using PATHway, and (5) Post programme reflection. There were a number of subthemes within each theme, for example motivation explores participants motivation to take part in PATHway and participants motivation to sustain engagement with PATHway throughout the intervention period. Participant engagement with the components of the PATHway system was variable. Future research should focus on optimising participant familiarisation with eHealth systems and employ an iterative approach to development and evaluation.

## Background

Globally, physical activity is a fundamental factor for the prevention of morbidity and mortality [1]. Specifically, lower levels of physical activity are associated with an increased incidence of cardiovascular disease (CVD) [2]. Cardiac rehabilitation (CR) is an exercise and education-

**Data Availability Statement:** All relevant data are within the paper and its Supporting Information files.

**Funding:** This project has received funding from the European Union's Horizon 2020 Framework Programme for Research and Innovation Action under Grant Agreement no. 643491:https://ec. europa.eu/digital-single-market/en/policies/ehealth This was awarded to KM, CW, NM, NMcC, WB, VC, and RB. No the funders had no role in study design, data collection and analysis, decision to publish, or preparation of the manuscript.

**Competing interests:** The authors have declared that no competing interests exist.

based programme for the secondary prevention of CVD and is associated with up to 26% reduction in cardiac mortality [3]. CR is defined in three stages including an inpatient phase, outpatient phase and a lifelong maintenance phase [4] However, despite the proven benefits, uptake and adherence to CR is low [5]. Researchers have explored the barriers to participation which include: a lack of access, a perceived lack of need, time and financial constraints, and a lack of individualised programmes [6, 7]. In the past decade there has been a focus on the development of interventions which use technology to overcome the barriers to attendance at centre-based CR programmes; these interventions appear to be as effective at increasing physical activity as centre-based programmes [8]. However, technology-based CR interventions are heterogeneous and have utilised several different components including, for example, biosensors, websites, mobile phones, fixed line phones in conjunction with exercise prescription, education, psychosocial support, behaviour change, text messaging and online tutorials (8). Much of the research examining eHealth CR interventions has mainly focused on the effectiveness by employing quantitative methods [8, 9]. Qualitative research enables us to gain an insight into users' views and experiences of the various components used in technology-based CR. This is required to expand our understanding of which components of an intervention are of most value to the user and how to improve their experience.

A mixed methods randomized controlled multicenter trial was undertaken to evaluate the acceptability, feasibility and clinical effectiveness of an eHealth phase 3 CR intervention: Physical Activity Towards Health (PATHway) [10]. PATHway is a personalized, lifestyle intervention which uses an integrated behaviour change approach with an internet-enabled and sensor-based home exercise platform as the core component. The aim of this study is to explore participants' views and experiences of using the PATHway system, an eHealth CR intervention.

## Methods

### Intervention

The PATHway-I trial took place between June 2016 and July 2018. PATHway was designed to enable participants to manage their CVD risk factors following discharge from outpatient CR programmes. PATHway was developed with patient and key stakeholder involvement through an iterative co-design process [11, 12]. Patients participating in outpatient CR were invited during the last four weeks of outpatient CR to take part in the trial. Written informed consent was obtained and participants were then randomized to either the intervention group (PATHway) or the control group (usual care) [10]. A detailed description of the PATHway-I trial and quantitative results can be found elsewhere [10, 13]. In brief, one hundred and twenty participants were recruited across three sites in two countries: The Mater Misericordiae University Hospital (Dublin), Beaumont Hospital (Dublin) and University Hospital Leuven (UZ Leuven, Belgium). Sixty participants were randomized to the intervention group and 60 to control group. Participants in the intervention group received the intervention (the PATHway system) for six months. The PATHway system is a home-based, technology enabled complex behavior change intervention. It provides regular exercise sessions as the basis upon which to provide a personalized, comprehensive lifestyle intervention program to enable patients to self-manage their CVD and to lead a healthier lifestyle in general. PATHway targeted specific lifestyle behaviors (diet, physical activity, smoking cessation, alcohol, stress reduction and medication adherence) which were personalized to each individual based on the results obtained from baseline assessments. A battery of outcome measures were used to assess these lifestyle behaviors Physical activity: Actigraph GT9X Link (worn for seven days), diet: Mediterrean Diet Score, alcohol: Alcohol Use Disorders test, stress: perceived stress scale and medication

adherence: Medication Adherence), full details of these are available in Claes et al. 2017 [10]. PATHway consists of 6 devices (portable PC including PATHway software, Microsoft Kinect camera, Microsoft Band 2 heart rate monitor, Blood pressure device, Zensor 3-lead ECG device, a headset) which enabled the delivery of the eleven components of the system: (1) Exer-Class: this component included dynamic aerobic and resistance exercises. Participant movements, repetition count, energy expenditure and heart rate (HR) were continuously monitored to provide personalised feedback via a virtual 'avatar' coach. (2) Screening: this component evaluated the participants resting HR and blood pressure, medication adherence and eating behaviour prior to engaging with ExerClass, this was utilised to help participants determine whether it was safe for them to engage in exercise. (3) Dashboard: combined data derived from ExerClass, ExerGame and outdoor physical activity was aggregated to generate a physical activity report, allowing participants to monitor their overall physical activity behaviour (4). Text messages: participants received text messages with information on reducing lifestyle-related risk factors for CVD. These text messages were tailored to the individual. Several lifestyle related cardiovascular risk factors were covered by PATHway: nutrition, stress, smoking, alcohol and medication adherence. Automated motivational physical activity messages were also sent congratulating participants on their activity levels, encouraging them to become more active or engaged, based on the activity recorded by the system. (5) Assessment: participants could self-assess their fitness with a two minute step test (6) ExerGame: this component provided participants with an opportunity to engage in game based exercise, the Microsoft Kinect sensor captured participants' movements as they were required to conduct certain exercises for example a squat, and their game avatar would jump on logs so as to cross a river without falling into water. (7) Instructions: this component contained detailed instructions on how to use the system. (8) Good Habits Visualisation (GHV): this 'Good Habits Visualisation' was personalised to each individual, based on the total scores from the lifestyle assessment completed by each individual at baseline. Their data was used to help them visualise how their current lifestyle fits in relation to CVD self-management guidelines and explores their willingness to change. Depending on their willingness to change participants could for example become involved in goal setting or be provided with educational support [13]. (9) Settings: the settings component enabled participants to alter the system to suit their needs, for example they could eliminate certain exercises from the ExerClass that aggravated their comorbidities or that they found uncomfortable. (10) Practice exercises: the practice exercises component allowed participants to isolate a certain exercise and practice it. (11) Calendar/events: to facilitate social support within a community context, the calendar/events component enabled small groups of remote participants to exercise together by allowing them to communicate during the exercise session. The calendar also allowed participants to promote events and to invite others to join. Additional detail on each of these components is available in (S1 Table). A demo video of the system and all its components is available online https://www.youtube.com/watch?v= FI39khtb0lg&t=148s. Ethics was obtained from the Ethics Committee UZ Leuven-KU Leuven and the Research Ethics Committee of both Irish hospital partners (KU Leuven: S59023; Mater Misericordiae Hospital Dublin: 1/378/1846; Beaumont Hospital Dublin: 16/50), as well as Dublin City University (DCU: REC2016/12). Written and informed consent was obtained from all participants. The consent procedures were approved by all the ethics committees involved.

Following randomisation, the PATHway system was installed in the participants' homes by a member of the research team. All participants were guided through how to use the equipment as well as each of the systems' 11 components (S1 Table) during a standardised familiarisation process of four sessions. During these sessions participants were systematically guided through how to use each piece of equipment and each of the eleven core components.

Participants then demonstrated competency with these at the subsequent sessions. Participants were notified of who to contact if they had any difficulty in operating the system.

## Data collection

Participant debriefs were conducted with all consenting participants who returned for post intervention testing at six months. The purpose and procedure for the debrief was explained to the participant at the start of each debrief and verbal consent obtained. In DCU debriefs were conducted in a quiet meeting room while in KUL debriefs took place in the participants home. Participants were initially shown a screen shot, one at a time, of each of the eleven components of PATHway to remind them of the component (S1 Table) and were asked: whether they used the element (if no, why not), how often they used it in a typical week and what they liked/disliked about it. Participants were then asked eight open ended questions regarding their views and experience of using the system (Table 1). Authors OOS and LMcD conducted the interviews in Dublin. Author RB conducted all interviews in Leuven. Interviews were audio recorded and transcribed verbatim.

## Data analysis

The interviews were analysed using Braun and Clarke [14] framework for thematic analysis. The steps involved in the framework are listed as follows:

1. Step one includes familiarising yourself with data through multiple readings.

2. Step two generates an initial list of ideas about what is in the data and what is interesting about them and involves the production of initial codes from the data.

3. Step three, themes begin to emerge, and this refocuses the analysis at the broader level of themes.

4. Step four involves reviewing themes whereby a set of candidate themes are explored and refined, including similarities and differences between interviews. This is an important step given the multisite approach in PATHway, which may offer conflicting findings.

5. Step five involves defining and naming themes

Steps 1 and 2 were conducted by authors OOS, LMcD and CW in Dublin and RB and NC in Leuven. The codes generated across the two sites in step 1 and 2 were then pooled together by OOS and presented to a senior member of the research team (CW). Steps 3 to 5 were then conducted in Dublin by OOS and CW. Provisional themes were generated and agreed upon by OOS and CW. Through an iterative process the themes were refined and re-defined until it was clear how they related to each other. All results were corroborated by authors from Leuven

**Table 1. PATHway debrief script, open ended questions.**

| |
|---|
| 1. What do you feel is the key contribution, if any, that PATHway has made to you? |
| 2. Tell us a little about how you used PATHway? |
| 3. What were some of the reasons that motivated you to used PATHway? |
| 4. What were some of the barriers you encountered when using PATHway? |
| 5. What were some of the strategies you used to overcome these barriers? |
| 6. Do you believe PATHway can be considered of equal value/effectiveness as the existing supervised group based phase 4 cardiac rehabilitation (for example Med-ex at DCU)? |
| 7. Can you give us three recommendations for change in order to improve PATHway in the future? |
| 8. Is there anything else you would like to add regarding your experience with PATHway? |

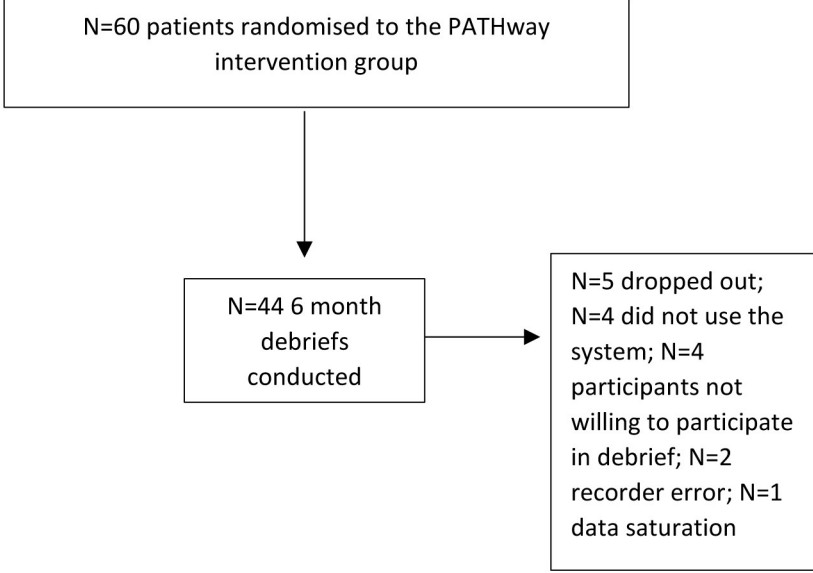

**Fig 1. Participant flow.**

to ensure that codes and representative quotes were integrated appropriately. Finally, quotes were selected to demonstrate each theme and the report was completed.

## Results

In total 44 debriefs were conducted, lasting a mean (standard deviation) of 23 (10) minutes. A flow diagram of participants is available in Fig 1, including reasons for non-participation in the debriefs. Thirty-four (77%) of the participants were male, full details of participants' characteristics are available in Table 2.

Five key themes were identified from the analysis of the participant debriefs: (1) Feedback on the components of the PATHway system, (2) Motivation, (3) Barriers to using PATHway, (4) Enablers to using PATHway, and (5) Post programme reflection. There were a number of subthemes within each of these themes.

### 1. Feedback on components of the PATHway system

Given that the PATHway trial was an acceptability and feasibility trial, it was of paramount importance to explore participants' thoughts on each of the eleven components, to help inform the further development of the PATHway system. Through this exploration it was identified that some components were used more than others and the reasons for this disparity in use were also identified. The usage and participant feedback on each component are summarised in Table 3.

Components such as the ExerClass, Screening, Dashboard and Text messages were individually reported to have the highest frequency of use and engagement. The ExerClass was viewed by some participants as the most important component of the PATHway system. Participants found it useful to be able to tailor the exercises and the length of the class; this allowed them to self-manage their exercises to suit their own time schedule. The Screening component was found to be reassuring for participants with regards to exercising safely, as well as providing a self-monitoring tool for their heart rate, blood pressure, medication adherence and eating behaviour. In relation to the Dashboard, some participants enjoyed being able to objectively

**Table 2. Characteristics of participants who completed a debrief.**

| Characteristic | Mean (SD) or frequency (n =) |
|---|---|
| Age **mean (SD)** | 61 (10) |
| Gender **M:F** | 34:10 |
| Reason for referral to cardiac rehabilitation | |
| Post PCI* | 14 |
| Post MI** | 15 |
| Post CABG*** | 9 |
| Post pacemaker | 3 |
| Post valve surgery | 1 |
| Unknown | 2 |
| Civil status ^ | |
| Married or living with a partner | 38 |
| Widowed | 2 |
| Single/divorced | 3 |
| Baseline moderate physical activity (minutes/day) **mean (SD)**^ | 125 (63) |
| Baseline daily step count **mean (SD)**^ | 12,575 (3,048) |

*PCI = percutaneous coronary intervention.

**MI = Myocardial infraction.

***CABG = Coronary artery bypass graph ^missing data n = 1.

see their level of activity and found it easy to use. Finally, nearly all participants reported receiving the Text Messages. While those participants who engaged with PATHway reported finding the messages motivational, some other participants reported that they did not always engage with them.

Instructions, ExerGame, Calendar/events, practice exercises, Settings and Good Habits Visualisation (GHV) appeared to have lower levels of engagement. Reasons for this included lack of awareness, perceived lack of need or lack of appeal. For example, despite each individual undergoing a standardised familiarisation protocol, which involved checking that they could use the components, some participants reported not being aware or having never seen the GHV, Settings or Practice Exercise components. Other participants reported being aware of these components but did not feel that they needed them. For example, in relation to the GHV component some participants felt that as they did not engage in the CVD risk related behaviours e.g. smoking, alcohol, high levels of stress then they only focused on physical activity and did not use GHV. In relation to the Calendar/Events a high number of participants reported that this did not meet their needs.

## 2. Motivation

Motivation was defined as participants' reasons for both initially taking part in PATHway and sustaining engagement with PATHway throughout the six-month intervention period.

**Participants' motivation for taking part in PATHway.** Varied and was mainly for health reasons, to continue to exercise with some level of support, and to continue to learn about their condition. A small number were influenced by the cardiac rehabilitation staff or felt the need to give back after the care they received.

"*For my health it is really necessary that I train, I know that I have to do it*" (Female, 40 years)

Table 3. Theme 1: Participant usage and feedback on the eleven components of PATHway.

| Component | Participants reporting using this component at least once (%) | Feedback on component |
|---|---|---|
| ExerClass | 95% | *"This really was the greatest added value of the entire system and very meaningful for me."* (Male, 50 years) *"What I liked about it was I was able to skip some of the exercises that I couldn't do. I was able to take a 10 minute session and then sometimes I'd actually just go and do another 10 minute session, because I felt I could do it. So I was able to manage my time that way."* (Male, 61 years) |
| Screening | 93% | *"Yes, I found that to be very great and very reassuring and kind of just nice to start off with, it kind of gave you a few reference numbers and you could kind of you know compare those with what they were before you know, so yes that bit I liked."*(Male, 47) *"I thought it was good to view and follow up on my own data."* (Male, 50 years) |
| Dashboard | 86% | *"I found this interesting to see an overview of my trainings here."* (Female, 40 years) *"Oh, it was simple and I was interested in the statistics. I spent a good bit of time on that app. I'd come back from a walk, for instance, checking distance, logging the stuff first and then going to the PATHway and checking."* (Male, 77 years) |
| Text messages | 80% | *"You reach your goal activity and they were good, they were good little motivators too. . . . . ., they focused the mind, even if I didn't do it at least it made me feel guilty about it, so it did encourage yeah."* (Male, 67 years) *"I received text messages for 'stress'. There were many and not always useful, but some messages did and I could use."* (Male, 60 years) |
| Assessment | 70% | "*Now this was not necessary because I already did bike tests*" (Female 66 years) *"It was actually, it was a very good, energetic test and actually would be a very interesting part of the programme, I found, that it's actually there so. . . . . . .. I only tried it twice. I just went back into, I went into using the programme then. So I don't.. I just thought, maybe I thought. . . it was part of the programme, there were step tests in the programme. I just felt they were more suited to what I was capable of, so."* (Male, 61 years) |
| ExerGame | 57% | *"I think it was quite nice, the one time I actually tried it with my grandson and he loved it. . . .."* (Male, 55 years) *"It didn't appeal to me, well maybe I'm just old fashioned but I don't think computers and exercise go together, you know. For me exercise is getting out of the house. . . .'* (Male, 54 years) |
| Instructions | 55% | *"Only in the beginning I really looked at it. once you know it, all instructions do not necessarily need to be shown every time"*(Female, 56 years) *"Didn't need it."* (Female, 50 years) |
| Good Habits Visualisation | 43% | *"I did not feel the need to use it . . . because my lifestyle was already in line with what is heart-friendly".* (Female, 64 years) *"I didn't know it was there."* (Female, 58 years) |
| Settings | 40% | *"I think I was shown it, but I didn't use it. I'd just skip on the programme. I'd just press the skip button if I didn't want to do the exercise. I'd press the skip button, so that was much easier to use."* (Male, 61 years) *"Never saw it"* (Female, 50 years) |

**Table 3.** (Continued)

| Component | Participants reporting using this component at least once (%) | Feedback on component |
|---|---|---|
| Practice exercises | 18% | "*I did not feel I needed this, because most exercises are the same as what we did during the rehabilitation*" (Female, 40 years)<br>"*Didn't do it, I didn't know it was there . . . I can't remember being shown it.*" (Male, 56 years) |
| Calendar/events | 5% | "*I looked at it, but I never used it . . . I don't think I actually set up a date to you know because my exercises are all at different times and it's when I get a chance to do it, usually it's in the evening but then it's all over the place.*" (Male, 55 years)<br>"*This is not necessary for me*" (Male, 74 years) |

"*To carry on the supervision of my condition, that I'd be monitored for an extra six months. I found it would be beneficial to me, to carry it on and give me a guide to what way to carry on in my life*" (Male, 62 years)

**Participants' motivation to sustain engagement with PATHway.** Was due to a range of factors. Self-monitoring of activity through wearing the Microsoft wrist band was motivational as participants could objectively see their results (for example how many calories they had burned) or monitor their daily activity (for example step count).

"*Yes. I'd check it when I'd be going around, to see what steps I was going on . . . I was getting about 17,000.*" (Male 62 years)

For others the text messages received as part of PATHway were motivational or they felt a sense of satisfaction from engaging with the system.

"*I used them all the time. Continuous, motivational texts, weekend, weekday, Bank Holidays, just to keep–very encouraging . . . and motivational.*" (Male, 61 years)

"*When you have done it, you are still happy.*" (Female, 58 years)

For a small number of participants knowing they were going to be tested again by the research team was a motivating factor.

"*. . . apart from the Pathway programme at home with the laptop, the three-monthly visits to DCU were, again, motivational and it inspired me because it gave me then a level of fitness. They were able to tell me how I was doing, and I thought that was brilliant.*" (Male, 61 years)

## 3. Barriers to using the PATHway system

Barriers to using the PATHway system describes problems that participants faced that either prevented or hampered their participation and/or performance of the PATHway system. Two subthemes were identified here: PATHway barriers and personal barriers.

**PATHway barriers.** PATHway barriers are defined as anything that specifically relates to the system which hampered a participant's engagement or use of the PATHway system. Some participants found elements of the PATHway system were unnecessarily complicated and that there were too many pieces of equipment. For example, participants reported that getting to complete the ExerClass took time as that they had to go through multiple steps before being

able to exercise, for example, turning on the system and then completing the Screening component (assessing whether it was safe to exercise).

> "*[screening]. . . was a bit cumbersome, it was another element to the whole start-up, so it was just another layer of . . . hassle.*" (Male, 45 years)

> "*I found that there was too many bits . . . you had a computer, you had the X-Box . . . and everything had to be plugged in a certain way and you didn't have enough power sockets. Then you kept running out of power and there were cables trailing. So it was a bit cumbersome. It wouldn't be something I'd set up myself.* "(Male, 61 years)

A number of participants experienced technical barriers in that the system did not work or parts of the system broke or were updating. The Microsoft band, and in particular its short battery life, was identified by participants as problematic.

> "*If I could not practice due to technical problems, I did not train anymore as I often planned to train in the evening. If I had lost a lot of time with technical problems, it was too late to train*" (Male, 66 years)

> "*It [*Microsoft *Band] didn't work on a couple of occasions,. . . so I found that very frustrating*". (Male, 56 years)

Finally, a small number of participants found that the exercises in the ExerClass were too strenuous and others found that the class lacked variation.

> "*Star jumps, after five minutes of doing them . . . you are nearly fit to quit.*" (Male, 54 years)

**Personal barriers.**   Personal barriers are defined as any barriers encountered by participants which were not directly related to the design of PATHway. A barrier to using the PATHway system for some participants was their low level of IT literacy.

> "*[I am] not computer literate. No knowledge of it [IT] at all.*" (Male, 75 years)

> "*The starting level was too high for me, I am 75 and I have never used a computer*" (Female, 75 years)

A number of participants had existing comorbidities with their CVD and for some this impacted on their use of the system. For example, one participant had rheumatoid arthritis which they felt limited their ability to engage with the system.

> "*I did not find the exercises unpleasant, although they were rather fierce on the legs and there was little variation*" (Female, 66 years)

> "*Sometimes my body was very painful . . . I didn't feel up to it at the time.*" (Male, 72 years)

Time is a frequently cited barrier to physical activity [15]. Even though PATHway was designed to provide flexibility in scheduling of exercise and physical activity, some participants had family or work commitments that still presented as a barrier to participation.

> "*I think I had one or two golden weeks and then like I had you know the odd time then for a while and then I just kind of gave it up because just work and everything*" (Male, 47 years)

"*I have little time because I am still working and also active as a musician*" (Female, 66 years)

For a small number of participants, bad weather and total lack of engagement presented as barriers.

"*I would get up in the morning and get the weather forecast and say 'oh, it's going to be a fine day, I'll go for a walk in the evening'. It could turn out in the evening it was lashing rain.*" (Male, 77 years)

"*Yeah, I didn't, it was . . . I'd no engagement with it whatsoever. The avatar, the fact it was a laptop, it was . . . you know, just didn't . . . didn't do it for me.*" (Male 45 years)

## 4. Enablers to using PATHway

Enablers refer to actions taken by participants or naturally occurring events that improved user engagement with the system and helped them overcome barriers. Two subthemes were identified under enablers: PATHway enablers and personal enablers.

**PATHway enablers.** PATHway enablers specifically relate to factors that improved engagement with the PATHway system or physical activity. When participants cited specific barriers to using the system, the interviewer asked how they overcame these. Technical support from the research team was available for all participants throughout the intervention and calling the PATHway team for IT support was a common enabler for participants when they encountered any technical barriers to using the system.

"*[Support Staff] came and fixed the system and got me back on the right track.*" (Male, 47 years)

The PATHway system was specifically designed to be individualised so that participants could tailor the time of day, mode and location (indoors/outdoors) of activity to meet their needs. This enabled participants to successfully complete their exercises and to overcome bad weather.

"*What I liked about it was I was able to skip some of the exercises that I couldn't do. And I was able to take a 10 minute session and then sometimes I'd actually just go and do another 10 minute session, because I felt I could do it. So I was able to manage my time that way. Sometimes I would do 20 minutes or sometimes I'd just do 10.*" (Male 61 years)

"*You know, so even when it was lashing rain or freezing cold outside you could exercise.*"(-Male 61 years)

**Personal enablers.** Personal enablers refer to factors outside of the PATHway system that enhanced participant performance; for example, some participants modified the exercises slightly to suit their personal ability.

"*I would slightly modify it, it would be the same intensity but I wouldn't be jumping as high and landing on the ground*" (Male, 65 years)

Finally, support from family members was a commonly cited enabler. For example some participants received IT support from family, while for others their physical activity/exercise was supported by family members.

"*My daughter is very good at IT. I would ask her and she would sort it out for me.*" (Male, 69 years)

"I*t was interesting that my wife also participated from the other side of the table, in her own way."* (Male, 74 years)

## 5. Post programme reflection

As part of the debriefs participants were asked: what contributions (if any) the system had made to them, their recommendations for improving the system, and if they felt that PATHway was equal to a structured and supervised cardiac rehabilitation class. Participants could also add any additional comments. The answers to these questions informed the theme of post programme reflection. The following subthemes were included: outcome evaluation, recommendations for improvement and whether PATHway was equal to a structured and supervised class.

**Outcome evaluation.**   Participants who engaged with the system felt the system had impacted on them by helping them to form a habit or routine around exercise.

"*Oh, it has developed a regime of exercise that I know I should do every day as a result of being in PATHway and it has reinforced that and reinforced diet.*"(Male, 69 years)

A high number of participants reported improvements in their health and fitness, including for example smoking cessation, pain reduction and improved strength

"*Yeah, I feel much better. [I am] off the cigarettes. . . walking up and down stairs. I think it has made me stronger. Before, walking the backs of my legs and all would be killing me, I just put that down to unhealthy, but it wasn't, it was the condition. Not getting that as much now.*" (Male, 47 years)

In general, most participants enjoyed utilising the PATHway system and participating in this research.

"*I thought it was very enjoyable. I enjoyed doing it, like you know what I mean.*" (Male 62 years)

Finally, a small number of participants reported improved self-efficacy; an improved ability to complete either every-day or recreational tasks.

"*It boosted me up, and if I can do that, I can do this. It was like last week, I think that's the first time we've. . . myself and X [a friend] have played a round of golf without a buggy in two years to two-and-a-half years. So, even though we were knackered after it, it was great to know we could do it.* "(Male, 70 years)

**Recommendations for improving PATHway.**   Participants' recommendations for improvement were varied but included reducing the number of pieces of equipment and improving the technical stability and the accuracy of heart rate measurements.

"*So, if it was a more portable version or that it plugged in to the television like an Xbox; I mean you could just use it or move it, it would be much better.*" (Male, 67 years)

"*Reliability must be increased both for technical stability and for heart rate measurements.*" (Male, 69 years)

Some participants reported they would have liked more variety in the exercises available in the ExerClass component.

"*It was all, overall like jumping jacks or kicking or stepping, and I couldn't do any of that. So I'd have preferred it if I could have sat in a chair and did some upper body exercises.*" (Male, 61 years)

"*Now all exercises are upright, but you can also do exercises while lying down and so on.*" (Male, 74 years)

Whilst the PATHway system can provide regular texts and emails, a small number of participants recommended tailoring the text messages to the individual's schedule as well as increasing feedback throughout the intervention (telehealth). Some recommended including music for the ExerClass.

"*Text messages should be more concrete, for instance you set your training schedule with dates and times and you get messages that it is time to train, that would be more helpful*" (Male, 66 years)

"*I think some sort of a regular email or text message . . . that provides more up to date information on your individual programme. Could be motivational as opposed to sort of generic bland messages that are trying to be motivational but didn't really achieve that goal.*" (Male, 48 years)

**Equal to a supervised and structured group class.** PATHway was developed to overcome barriers to attendance and adherence at supervised and structured community-based programmes. Participants had mixed feelings as to whether the PATHway system was equal to a structured supervised phase 3, community-based, CR class. A high number felt that the flexibility of the system was very beneficial, and it had the potential to be as effective.

"*Yes. I do definitely . . . because time wise. Personally I found the class in [site of phase 2 cardiac rehabilitation] brilliant, but I was struggling to get there time wise. Physically it would be very hard for me to commit to doing it, but PATHway, certainly at home if you had your own system of PATHway at home, absolutely it would be as good.*" (Male, 56 years)

Other participants felt it was down to an individual preference, whether someone preferred to exercise independently or as part of a group in a structured supervised environment. While some participants felt that PATHway had potential to be equal to a supervised and structured class if the technology was improved.

"*That really depends on individuals. Some people like to work in groups and get their strong motivation from having to come down and join the group and do it.*" (Male, 67 years)

"*If the technical aspects had been in order, it could be good to keep your condition up to date*" (Female, 66 years)

Finally, some participants felt that PATHway was lacking both supervision from an expert as well as direct group contact.

*"I missed the personal contact with the care providers, for me it was not enough help to get me started with this because I never really used a computer before."* (Female, 74 years)

"*Being on your own operating a software system for exercise certainly has disadvantages compared to being in a group who have a supervisor–and when you have kind of an overseer watching what everybody is doing, that's an advantage I think.*" (Male, 57 years)

## Discussion

This study explored participants' views and experiences of using an eHealth cardiac rehabilitation platform, PATHway. Participant engagement with PATHway was variable. Some components were engaged with more frequently than others. Considering the results of this qualitative exploration in the context of the wider literature can help us gain an understanding of this variation and provide us with important learning for the future development of PATHway and evaluation of future eHealth interventions.

The PATHway intervention is a behaviour change intervention, which used an exercise platform to provide a personalized, comprehensive lifestyle intervention. Specific behaviour change techniques were utilised including: self-monitoring, feedback, social support and individual tailoring [16] and these appeared to be integral to participants motivation to initiate and to sustain engagement in addition to enabling them to overcome barriers. The PATHway components with the highest level of self-reported engagement were ExerClass, Screening and the Dashboard, each of these allowed for self-monitoring of physical activity and health related parameters, including blood pressure and heart rate and provided participants with feedback on these. Self-monitoring has been shown to be an effective tool in improving physical activity in CVD patients [17] and other populations [18, 19]. With improvements in technology, self-monitoring of health status is becoming increasingly used in everyday life [20]. Interestingly a motivation for partaking in the current trial for some participants was to continue to exercise with some degree of safety and support. Therefore, providing patients who are post phase 2 CR with a means to monitor their physical activity and health status, along with appropriate education, may facilitate more sustained engagement in a healthier lifestyle.

This qualitative exploration demonstrated that components such as Good Habits Visualisation, Settings, Practice Exercises and Calendar/Events were used infrequently; with some participants reporting that they were not aware of them and/or did not need them. This lack of awareness may seem surprising given that each participant was exposed to a standardised four-week familiarisation period. Previous qualitative research has found that participants felt they needed more than a familiarization period before they would feel confident about using the technology [19], especially when a system encompasses a wide range of services as was the case with PATHway. It would seem there is a need for dedicated research into the appropriate strategies needed in eHealth interventions to optimise participants' familiarity with all components of the system. This has the capacity to overcome some of the barriers reported in the current study, including for example awareness of the settings component which would have allowed users to remove exercises that they found difficult or uncomfortable. Furthermore, in an effort to optimise user engagement and acceptability PATHway was informed by extensive formative research and incorporated a co-design approach which involved key stakeholders throughout the development process [12]. Despite this, participants reported a perceived lack of need with some components, specifically in relation to the Good Habits Visualisation and the Calendar/Events. The perceived lack of need of the Good Habits Visualisation is potentially due to a lack of understanding or poor perception of CVD risk and not a lack of motivation to

change as participants reported health as a motivation to engage with PATHway in the first instance; current research indicates that patients tend to generally underestimate their CVD risk [21]. All participants had received structured education regarding CVD risk factors during outpatient CR prior to participating in the trial. Therefore, participants might have felt they sufficiently implemented behaviour changes towards a healthier lifestyle already, as demonstrated in their baseline physical activity (Table 2). Despite this, perhaps additional reinforcement and objective measures of individual risk (e.g. food diaries for nutrition) are required to overcome this perceived lack of need as there was room for further improvement [13]. It could also be argued that the delivery of education in a group setting of cardiac rehabilitation during outpatient CR does not adequately meet each individuals' needs and preferences, although previous research in diabetes patients has shown equal effectiveness between group and individual education [22]. The Calendar/Events was the mechanism by which participants could initiate a group activity (e.g. meeting up for a walk, completing an ExerClass as a small group). The reported lack of need of Calendar/Events may be justified by the availability of social support from family and friends as evidenced in the enablers reported. However, some participants reported that they missed having the group element of the outpatient CR phase, and a suggestion for improvement of the system was to incorporate a group element. Future work is required to engage with participants during the familiarisation stages to identify specifically what their needs are and to match the intervention components to the individual.

Finally, in addition to the variation in engagement there was a lack of uniformity in participants' responses for how they felt PATHway could be improved and whether it was equal to a structured and supervised exercise class. This demonstrates how individual a person's needs are in relation to physical activity and exercise. Perhaps the expectation of individuals to use 'all' elements of a technology intervention is false, rather research needs to continually observe use of the 'innovation' in practice to determine which components are used and the rationale for engagement [23]. Behaviour change research suggests that interventions should include a *sufficient* number of behaviour change techniques to enable participants to select those which best suit their needs [24]; findings from the current study would indicate the same is true for eHealth intervention components. Therefore, the development of future complex eHealth interventions should include more regular surveillance and monitoring of participant engagement with the components of the intervention [25] to allow for a more iterative approach to the evaluation of eHealth interventions [26]. Ultimately greater understanding of individuals' preferences and identifying phenotypes could help better stratify participant engagement with eHealth interventions or community-based interventions.

While the current exploration of participants' views and experiences of the PATHway system is novel and provides us with valuable insights from participants in two different countries, it is not without its limitations. Firstly, the semi structured interviews were conducted in two different countries by different researchers, in different languages, while procedures were put in place to try and ensure rigour and reduce variability across sites, it is possible that some of cultural context of the quotes was lost. It is important to acknowledge that the current qualitative analysis included data from participants who did not use the system beyond the familiarisation stage, the inclusion of these participants' views was considered important as usability is not all or nothing but is a continuum and these participants' views provide us with additional insights into PATHway. We did not capture views of participants who dropped out, we therefore do not know what kind of experience these participants had. Finally, given the feasibility nature of the PATHway trial there were issues with the technology in the earlier stages of the intervention which presented as barriers to participants and may have reduced participants engagement with the system as well as their views of the system.

## Conclusion

The PATHway system provided participants with a platform to engage in physical activity and exercise within a safe and structured format. Behaviour change techniques including self-monitoring, feedback, social support and tailoring appeared to be useful to promote engagement with the system. However, not all participants engaged with the PATHway system and not all components were engaged with fully. Future research evaluating eHealth interventions should seek to continuously monitor participant engagement with components and allow individuals to tailor a package to meet their specific needs.

## Supporting information

**S1 Table. Description of the eleven PATHway components.**
(DOCX)

## Author Contributions

**Conceptualization:** Catherine Woods, Roselien Buys, Véronique Cornelissen, Niall Moyna, Noel McCaffrey, Werner Budts, Kieran Moran.

**Data curation:** Roselien Buys, Véronique Cornelissen, Kieran Moran.

**Formal analysis:** Orlagh O'Shea, Catherine Woods, Lauri McDermott, Roselien Buys, Nils Cornelis, Jomme Claes, Véronique Cornelissen, Clare McDermott, Ciara McCormack, Kieran Moran.

**Funding acquisition:** Catherine Woods, Roselien Buys, Véronique Cornelissen, Niall Moyna, Noel McCaffrey, Werner Budts, Kieran Moran.

**Investigation:** Orlagh O'Shea, Lauri McDermott, Roselien Buys, Nils Cornelis, Jomme Claes, Véronique Cornelissen, Anne Gallagher, Helen Newton, Noel McCaffrey, Davide Susta, Clare McDermott, Ciara McCormack, Kieran Moran.

**Methodology:** Orlagh O'Shea, Catherine Woods, Lauri McDermott, Roselien Buys, Nils Cornelis, Jomme Claes, Véronique Cornelissen, Niall Moyna, Noel McCaffrey, Clare McDermott, Ciara McCormack, Werner Budts, Kieran Moran.

**Project administration:** Orlagh O'Shea, Catherine Woods, Lauri McDermott, Roselien Buys, Nils Cornelis, Jomme Claes, Véronique Cornelissen, Anne Gallagher, Helen Newton, Davide Susta, Clare McDermott, Ciara McCormack, Kieran Moran.

**Resources:** Véronique Cornelissen, Kieran Moran.

**Supervision:** Lauri McDermott, Roselien Buys, Véronique Cornelissen, Anne Gallagher, Helen Newton, Niall Moyna, Noel McCaffrey, Davide Susta, Werner Budts, Kieran Moran.

**Validation:** Orlagh O'Shea, Catherine Woods, Lauri McDermott, Roselien Buys, Véronique Cornelissen, Kieran Moran.

**Visualization:** Catherine Woods, Kieran Moran.

**Writing – original draft:** Orlagh O'Shea, Catherine Woods, Kieran Moran.

**Writing – review & editing:** Orlagh O'Shea, Catherine Woods, Lauri McDermott, Roselien Buys, Nils Cornelis, Jomme Claes, Véronique Cornelissen, Anne Gallagher, Helen Newton, Niall Moyna, Noel McCaffrey, Davide Susta, Clare McDermott, Ciara McCormack, Werner Budts, Kieran Moran.

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
