## [Decision Letter · Decision Letter 0]

4 Feb 2020

PONE-D-20-00433

A qualitative exploration of cardiovascular disease patients’ views and experiences with an eHealth cardiac rehabilitation intervention: The PATHway Project

PLOS ONE

Dear Dr O'Shea,

Thank you for submitting your manuscript to PLOS ONE. After careful consideration, we feel that it has merit but does not fully meet PLOS ONE’s publication criteria as it currently stands. Therefore, we invite you to submit a revised version of the manuscript that addresses the points raised during the review process.

We would appreciate receiving your revised manuscript by Mar 20 2020 11:59PM. To enhance the reproducibility of your results, we recommend that if applicable you deposit your laboratory protocols in protocols.io, where a protocol can be assigned its own identifier (DOI) such that it can be cited independently in the future. For instructions see: http://journals.plos.org/plosone/s/submission-guidelines#loc-laboratory-protocols

We look forward to receiving your revised manuscript.

Kind regards,

Wen-Jun Tu

Academic Editor

PLOS ONE

2. Please amend your current ethics statement to address the following concerns: Please explain why was written consent was not obtained, how you recorded/documented participant consent, and if the ethics committees/IRBs approved this consent procedure.

Reviewers' comments:

Reviewer's Responses to Questions

**Comments to the Author**

1. Is the manuscript technically sound, and do the data support the conclusions?

Reviewer #1: Yes

Reviewer #2: Yes

2. Has the statistical analysis been performed appropriately and rigorously? 

Reviewer #1: N/A

Reviewer #2: N/A

3. Have the authors made all data underlying the findings in their manuscript fully available?

Reviewer #1: Yes

Reviewer #2: Yes

4. Is the manuscript presented in an intelligible fashion and written in standard English?

Reviewer #1: Yes

Reviewer #2: Yes

5. Review Comments to the Author

Reviewer #1: The aim of this study was to explore participants’ views and experiences of an eHealth phase 3

cardiac rehabilitation (CR) intervention: Physical Activity Towards Health (PATHway) and next

Authors write “lifestyle behaviors (diet, physical activity, smoking cessation, alcohol, stress reduction and medication adherence) which were personalized to each individual based on the results obtained from baseline assessments” or in another place “participants received text messages with information on reducing lifestyle-related risk factors for CVD. These text messages were tailored to the individual. Several lifestyle related cardiovascular risk factors were covered by PATHway: nutrition, stress, smoking, alcohol and medication adherence” and present Pathway as a system encompassing several health behaviors, but the research/results concern(s) actually physical activity only and no other health behaviors.

And it was not clearly for me what the assessments looked like? In what way they were personalized?

(p.5)

“Participant debriefs were conducted with all consenting participants who returned for post

intervention testing at six months” – I wonder what happened with participants who did not return and why they did not returned; what sort of experiences they had? (p.8)

“At its core, the PATHway intervention is a behaviour change intervention for physical activity, diet, smoking cessation, medication adherence and stress” and next “The PATHway components with the highest level of self-reported engagement were ExerClass, Screening and the Dashboard, each of these allowed for self-monitoring of physical activity and health related parameters, including blood pressure and heart rate and provided participants with feedback on these” – what about the rest of health behaviors mentioned here and above? (p.22)

Please consider more about limitations of the research, e.g. about limitations of applied metodological approach.

p.4 - once “60” and ones “sixty” – my suggestion is to apply APA guidelines

Reviewer #2: The authors report on a qualitative study exploring patients´ views and experiences of using an eHealth cardiac rehabilitation intervention (The PATHway Project). This manuscript is part of a series of papers reporting data from the same trial.

1) Page 3. What is meant with “allows us to adopt a more person centred approach”? Do you mean that qualitative research helps to tailor interventions based on each patients´ experiences, situation and needs?

2) Page 4. What was the rationale for inviting patients during the last four weeks of outpatient CR?

3) Different phases of CR are mentioned in the paper (phase 2, 3 and 4). How are these phases defined?

4) Participants included have a mean age of 61 and 77% were men. Representative? Was a specific group of patients more attracted to participate and use the Pathway system?

5) Where were the debriefs conducted? (at a CR unit). How long did they take? (mean, median and range). Richness in data of each debrief?

6) Any differences between the codes generated at the two sites? Was the codes generated in Leuven translated into English? How?

7) Physical activity seem to be of priority in the Pathway system. However, it was also designed to target other life style factors (nutrition, smoking, stress, alcohol, medication adherence). How the system is supposed to work to, e.g. help participants quit smoking or reduce alcohol consumption, can be further clarified.

8) I suggest considering adding the participants levels of physical activity in table 1. These are presented in REF 10, but the analysis of the debriefs in the present study, may be reflected in light of PA levels (see further my comment 13). Information on civil status can also be added in table 1.

9) Page 5, line 10. I suggest replacing compliance with adherence.

10) The authors suggest using an iterate approach on optimizing participants’ familiarization with eHealth systems. Such an iterative co-design was performed in this project (also described in REF 12), and the participants were systematically guided through how to use each component of the system, but still the engagement with the components in the system was variable. Do you consider the iterative approach that was used in this study as successful? How can it be improved?

11) Ref 25 is not complete. Dou you mean this REF?

Moore GF, Audrey S, Barker M, Bond L, Bonell C, Hardeman W, et al. Process evaluation of complex interventions: Medical Research Council guidance. BMJ : British Medical Journal. 2015;350:h1258.

12) Figure 1. Given that 44 debriefs were conducted, what is meant with that one participant was excluded due to “data saturation”?

13) I think it was a well considered choice by the authors to include participants who did not use the system beyond the familiarisation stage to get a more nuanced picture. Although the participants were equipped with advanced support to implement life style changes this paper illustrates the challenges with implementing life style changes. Do you consider the intervention group as motivated to perform lifestyle changes and as a group with a high CVD risk factor profile (i.e. patients in most need of performing life style changes)? A discussion in light of that may strengthen the paper.

6. PLOS authors have the option to publish the peer review history of their article (what does this mean?). If published, this will include your full peer review and any attached files.

Reviewer #1: No

Reviewer #2: No

---

## [Author Response · Author response to Decision Letter 0]

20 Mar 2020

Dear Reviewers,

We thank you for taking the time to review this manuscript and for your comments. We have addressed all of the comments below and where appropriate made changes to the manuscript.

Kind regards,

Orlagh O’Shea

Reviewer 1

The aim of this study was to explore participants’ views and experiences of an eHealth phase 3 cardiac rehabilitation (CR) intervention: Physical Activity Towards Health (PATHway) and next Authors write “lifestyle behaviors (diet, physical activity, smoking cessation, alcohol, stress reduction and medication adherence) which were personalized to each individual based on the results obtained from baseline assessments” or in another place “participants received text messages with information on reducing lifestyle-related risk factors for CVD. These text messages were tailored to the individual. Several lifestyle related cardiovascular risk factors were covered by PATHway: nutrition, stress, smoking, alcohol and medication adherence” and present Pathway as a system encompassing several health behaviors, but the research/results concern(s) actually physical activity only and no other health behaviors. And it was not clearly for me what the assessments looked like? In what way they were personalized?

(p.5) 

PATHway is described as complex behavior change intervention which uses an exercise platform as the core component but also addresses lifestyle behaviors (diet, smoking cessation, alcohol, stress reduction and medication adherence). The results mainly concern the physical activity/exercise behavior as this was the core component and the component that participants engaged with the most. Other lifestyle behaviors were addressed through the “Good Habits Visualisation” health behavior change component which was personalized to each individual based on their baseline assessments. It is detailed in the results that participants did not engage with the “Good Habits Visualisation” either because they felt they did not need it or they were not aware of it. Both of these issues are addressed in the discussion. 

Additional wording has been added to emphasise that exercise was the core component of PATHway and to detail the baseline assessment of the lifestyle behaviors. Page 5 105-107, 110-113 and Page 6 133-137

“Participant debriefs were conducted with all consenting participants who returned for post intervention testing at six months” – I wonder what happened with participants who did not return and why they did not returned; what sort of experiences they had? (p.8) 

Reasons for not returning to 6 month assessments have been provided in, reference 14: Claes et al. J Med Internet Res 2020;22(2):e14221) and included serious adverse events, loss of interest, onset of disease with exercise contraindication or mental health issues. Unfortunately we did not capture these participants’ views and this has been commented on in the limitations. 

“At its core, the PATHway intervention is a behaviour change intervention for physical activity, diet, smoking cessation, medication adherence and stress” and next “The PATHway components with the highest level of self-reported engagement were ExerClass, Screening and the Dashboard, each of these allowed for self-monitoring of physical activity and health related parameters, including blood pressure and heart rate and provided participants with feedback on these” – what about the rest of health behaviors mentioned here and above? (p.22)

We have amended this sentence, as the reviewer correctly points out that the emphasis of our results and manuscript were on physical activity, page 23 283. 

Please consider more about limitations of the research, e.g. about limitations of applied methodological approach. Recognition of further limitations have been added page 25 346-355. once “60” and ones “sixty” – my suggestion is to apply APA guidelines 

This manuscript was written in line with APA guidelines, in this instance “sixty,” was written in words as it was at the start of a sentence. 

Reviewer 2

 1)Page 3. What is meant with “allows us to adopt a more person centred approach”? Do you mean that qualitative research helps to tailor interventions based on each patients´ experiences, situation and needs? 

The text has been amended to address any ambiguity with regard to this statement. Please see highlighted text on page 3-4, 79-85.

“Qualitative research enables us to gain an insight into users’ views and experiences of the various components used in technology-based CR. This is required to expand our understanding of which components of an intervention are of most value to the user and how to improve their experience. While quantitative data gives us information with regard to the effectiveness of the intervention it does not tell us whether the participants actually enjoyed or liked the intervention has been documented predictor of maintenance (10).”

2) Page 4. What was the rationale for inviting patients during the last four weeks of outpatient CR? 

The rationale for inviting people during the last 4 weeks of rehab was a pragmatic choice: It was to conduct the familiarization phase of PATHway whilst they were concluding phase 3, in this way they could experience the intervention, the technology, complete assignments and ask any questions to the researcher over the familarisation period. 

3) Different phases of CR are mentioned in the paper (phase 2, 3 and 4). How are these phases defined? 

The phases of cardiac rehabilitation are defined as follows: CR is traditionally divided into three phases. Phase I is typically an inpatient service, which consists of early mobilisation, brief counselling about the nature of the illness, the treatment, risk factor management and follow-up planning. Phase II is mainly a supervised ambulatory outpatient programme. Phase III is a lifetime maintenance phase where the goal is to continue the risk factor- and lifestyle change and exercise training.

An additional sentence and reference have been included to address this page 3 67-68. 

4) Participants included have a mean age of 61 and 77% were men. Representative? Was a specific group of patients more attracted to participate and use the Pathway system? 

A comparison between eligible consenting (n=120) and nonconsenting participants (n=98) revealed a significant difference in age, with older

participants being less likely to enrol in the study (60.3 [SD 9.2] years vs 64.7 [SD 9.2 years]; P=.001). This is detailed in reference 14: Claes et . J Med Internet Res 2020;22(2):e14221)

Additionally, there is a higher incidence of cardiovascular disease in men and current literature indicates the mean age of participants engaging in cardiac rehab is 64 years and 70% of these are men (Gaalema et al. 2019). 

Gaalema, D.E., Savage, P.D., Leadholm, K, Rengo, J., Naud, S., Priest, J.S., Ades, P.A. (2019). Clinical and Demographic Trends in Cardiac Rehabilitation: 1996-2015. Journal of Cardiopulmonary Rehabilitation and Prevention; 39:266-273

5) Where were the debriefs conducted? (at a CR unit). How long did they take? (mean, median and range). Richness in data of each debrief? 

The debriefs were conducted in a quiet meeting room at DCU at the 6 month testing in KUL At KUL, the debriefs were conducted at the home of the participants, as the debrief was combined with the collection of the PATHway equipment. Debriefs lasted a mean (standard deviation) of 23 (10) minutes. This detail has been added to page 8 line 10-12 and age 10 line 42 respectively.

6) Any differences between the codes generated at the two sites? Was the codes generated in Leuven translated into English? How? 

There was no difference in themes between the two sites. The data from KUL was translated into English by the KUL team who are fluent in English speakers.

7) Physical activity seem to be of priority in the Pathway system. However, it was also designed to target other life style factors (nutrition, smoking, stress, alcohol, medication adherence). How the system is supposed to work to, e.g. help participants quit smoking or reduce alcohol consumption, can be further clarified. 

At baseline these health behaviors were assessed and then the “Good Habits Visualisation” component was personalized to them, to communicate which behaviors required some changes in line with CVD self-management guidelines. Participants could choose a health behavior which they wanted to change and the system guided them through this based on their readiness to change for example some participants would have been set a goal or others would have been provided with some educational support. Full details of this are available in reference (13). Changes have been made to the text to reflect this (page 6 133-138).

8) I suggest considering adding the participants levels of physical activity in table 1. These are presented in REF 10, but the analysis of the debriefs in the present study, may be reflected in light of PA levels (see further my comment 13). Information on civil status can also be added in table 1 

Information on participants data captured by PATHway system was anonymized, therefore we do not have their physical activity levels during the intervention.

9) Page 5, line 10. I suggest replacing compliance with adherence 

This has been done. 

10) The authors suggest using an iterate approach on optimizing participants’ familiarization with eHealth systems. Such an iterative co-design was performed in this project (also described in REF 12), and the participants were systematically guided through how to use each component of the system, but still the engagement with the components in the system was variable. Do you consider the iterative approach that was used in this study as successful? How can it be improved? 

An iterative process was employed as part of the design of this intervention (reference 12), which appears to have been successful. The iterative approach was not continued during the intervention testing period which may have resulted in variables level of engagement, thus the author suggest using an iterative approach for optimising participant engagement during the intervention period. Further clarification has been added to this paragraph page 24 339-344. 

11) Ref 25 is not complete. Do you mean this REF?

Moore GF, Audrey S, Barker M, Bond L, Bonell C, Hardeman W, et al. Process evaluation of complex interventions: Medical Research Council guidance. BMJ : British Medical Journal. 2015;350:h1258.

 Thank you for this recommendation this has been updated.

12) Figure 1. Given that 44 debriefs were conducted, what is meant with that one participant was excluded due to “data saturation”? Data saturation refers to the point in the research process when no new information is discovered in data analysis. Researchers in KUL conducting the semi structured interviews did not conduct one interview as they felt that they were not discovering any new information from the interviews. 

13) I think it was a well considered choice by the authors to include participants who did not use the system beyond the familiarisation stage to get a more nuanced picture. Although the participants were equipped with advanced support to implement life style changes this paper illustrates the challenges with implementing life style changes. Do you consider the intervention group as motivated to perform lifestyle changes and as a group with a high CVD risk factor profile (i.e. patients in most need of performing life style changes)? A discussion in light of that may strengthen the paper The authors consider the participants in the current study motivated to perform lifestyle changes as a theme that emerged in this study was that health was a key reason for engaging with PATHway in the first instance. However as outlined in the discussion CVD patients can underestimate their CVD risks which potentially manifested as a lack of engagement with some components, additional wording has been added in an attempt to clarify this page 23, 314-315.

---

## [Decision Letter · Decision Letter 1]

9 Apr 2020

PONE-D-20-00433R1

A qualitative exploration of cardiovascular disease patients’ views and experiences with an eHealth cardiac rehabilitation intervention: The PATHway Project

PLOS ONE

Dear Dr O'Shea,

Thank you for submitting your manuscript to PLOS ONE. After careful consideration, we feel that it has merit but does not fully meet PLOS ONE’s publication criteria as it currently stands. Therefore, we invite you to submit a revised version of the manuscript that addresses the points raised during the review process.

We would appreciate receiving your revised manuscript by May 24 2020 11:59PM. To enhance the reproducibility of your results, we recommend that if applicable you deposit your laboratory protocols in protocols.io, where a protocol can be assigned its own identifier (DOI) such that it can be cited independently in the future. For instructions see: http://journals.plos.org/plosone/s/submission-guidelines#loc-laboratory-protocols

We look forward to receiving your revised manuscript.

Kind regards,

Wen-Jun Tu

Academic Editor

PLOS ONE

Reviewers' comments:

Reviewer's Responses to Questions

**Comments to the Author**

1. If the authors have adequately addressed your comments raised in a previous round of review and you feel that this manuscript is now acceptable for publication, you may indicate that here to bypass the “Comments to the Author” section, enter your conflict of interest statement in the “Confidential to Editor” section, and submit your "Accept" recommendation.

Reviewer #2: (No Response)

2. Is the manuscript technically sound, and do the data support the conclusions?

Reviewer #2: Yes

3. Has the statistical analysis been performed appropriately and rigorously? 

Reviewer #2: N/A

4. Have the authors made all data underlying the findings in their manuscript fully available?

Reviewer #2: Yes

5. Is the manuscript presented in an intelligible fashion and written in standard English?

Reviewer #2: Yes

6. Review Comments to the Author

Reviewer #2: I have re-reviewed this manuscript, which has improved but I have some minor comments.

1) Page 4. I suggest to delete the new sentence “While quantitative data gives us information with regard to the effectiveness of the intervention it does not tell us whether the participants actually enjoyed or liked the intervention has been documented predictor of maintenance (10)”.

I think this sentence is not needed as the sentence before (regarding qualitative) has been clarified.

2) Below is from our previous conversation:

I suggest considering adding the participants levels of physical activity in table 1.

These are presented in REF 10, but the analysis of the debriefs in the present study,

may be reflected in light of PA levels (see further my comment 13). Information on civil status can also be added in table 1

Authors´ response

Information on participants data captured by PATHway system was anonymized,

therefore we do not have their physical activity levels during the intervention.

Additional comment: My suggestion was to add baseline PA levels and civil status in Table 2.

3) Page 23 Referring to this sentence “At its core, the PATHway intervention is a behaviour change intervention...”

Here it can be stated that the exercise platform is the core component in the PATHway system (to be in line with other changes performed in the method section).

4) Page 25. Regarding the following sentence in the discussion:

“The perceived lack of need of the Good Habits Visualisation is potentially due to a lack of understanding or poor perception of CVD risk and not a lack of motivation to change as participants reported health as a motivation to engage with PATHway in the first instance”.

Additional comment: Can it be the case that professionals fails to meet patients as persons with individual preferences, needs and resources (not tailored for each person)?, and which may lead to a lack of understanding. I suggest to change/elaborate the sentence above.

7. PLOS authors have the option to publish the peer review history of their article (what does this mean?). If published, this will include your full peer review and any attached files.

Reviewer #2: No

---

## [Author Response · Author response to Decision Letter 1]

17 May 2020

1) Page 4. I suggest to delete the new sentence “While quantitative data gives us information with regard to the effectiveness of the intervention it does not tell us whether the participants actually enjoyed or liked the intervention has been documented predictor of maintenance (10)”.

I think this sentence is not needed as the sentence before (regarding qualitative) has been clarified.

Response: Thank you for this recommendation this sentence has been removed.

2) Below is from our previous conversation:

I suggest considering adding the participants levels of physical activity in table 1.

These are presented in REF 10, but the analysis of the debriefs in the present study,

may be reflected in light of PA levels (see further my comment 13). Information on civil status can also be added in table 1

Authors´ response

Information on participants data captured by PATHway system was anonymized,

therefore we do not have their physical activity levels during the intervention.

Additional comment: My suggestion was to add baseline PA levels and civil status in Table 2.

Response: Baseline physical activity and civil status have been added to Table 2

3) Page 23 Referring to this sentence “At its core, the PATHway intervention is a behaviour change intervention...”

Here it can be stated that the exercise platform is the core component in the PATHway system (to be in line with other changes performed in the method section).

Response: In the methods PATHway is described as “a home-based, technology enabled complex behavior change intervention. It provides regular exercise sessions as the basis upon which to provide a personalized, comprehensive lifestyle intervention program to enable patients to self-manage their CVD and to lead a healthier lifestyle in general.” It was designed for the exercise to be the core component. A minor change has been made to this sentence (page 24, line 286-288):

 The PATHway intervention is a behaviour change intervention and specific behaviour change techniques were utilised including: self-monitoring, feedback, social support and individual tailoring (16)

4) Page 25. Regarding the following sentence in the discussion:

“The perceived lack of need of the Good Habits Visualisation is potentially due to a lack of understanding or poor perception of CVD risk and not a lack of motivation to change as participants reported health as a motivation to engage with PATHway in the first instance”.

Additional comment: Can it be the case that professionals fails to meet patients as persons with individual preferences, needs and resources (not tailored for each person)?, and which may lead to a lack of understanding. I suggest to change/elaborate the sentence above.

Response: Thank for you for this comment we have made changes to reflect the lack of personalisation in the group setting out cardiac rehabilitation. See page 25 line 325- 328.

It could also be argued that the delivery of education in a group setting of cardiac rehabilitation during outpatient CR does not adequately meet each individuals’ needs and preferences, although previous research in diabetes patients has shown equal effectiveness between group and individual education (22).

---

## [Decision Letter · Decision Letter 2]

12 Jun 2020

A qualitative exploration of cardiovascular disease patients’ views and experiences with an eHealth cardiac rehabilitation intervention: The PATHway Project

PONE-D-20-00433R2

Dear Dr. O'Shea,

We’re pleased to inform you that your manuscript has been judged scientifically suitable for publication and will be formally accepted for publication once it meets all outstanding technical requirements.

Kind regards,

Wen-Jun Tu

Academic Editor

PLOS ONE

Additional Editor Comments (optional):

Reviewers' comments:

Reviewer's Responses to Questions

**Comments to the Author**

1. If the authors have adequately addressed your comments raised in a previous round of review and you feel that this manuscript is now acceptable for publication, you may indicate that here to bypass the “Comments to the Author” section, enter your conflict of interest statement in the “Confidential to Editor” section, and submit your "Accept" recommendation.

Reviewer #2: (No Response)

2. Is the manuscript technically sound, and do the data support the conclusions?

Reviewer #2: Yes

3. Has the statistical analysis been performed appropriately and rigorously? 

Reviewer #2: N/A

4. Have the authors made all data underlying the findings in their manuscript fully available?

Reviewer #2: Yes

5. Is the manuscript presented in an intelligible fashion and written in standard English?

Reviewer #2: Yes

6. Review Comments to the Author

Reviewer #2: I am satisfied that all my previous comments have been adequately addressed.

1)In Table 1, civil status is missing for one participant. Add or indicate as missing.

2)In the reference list there are two references with number 24. Check and update.

7. PLOS authors have the option to publish the peer review history of their article (what does this mean?). If published, this will include your full peer review and any attached files.

Reviewer #2: No

---

## [Editor Report · Acceptance letter]

16 Jun 2020

PONE-D-20-00433R2 

A qualitative exploration of cardiovascular disease patients’ views and experiences with an eHealth cardiac rehabilitation intervention: The PATHway Project 

Dear Dr. O'Shea:

I'm pleased to inform you that your manuscript has been deemed suitable for publication in PLOS ONE. Congratulations! Your manuscript is now with our production department. 

Kind regards, 

on behalf of

Dr. Wen-Jun Tu 

Academic Editor

PLOS ONE